# Evaluation of the Cell Block Method Using Overnight-Stored Bile for Malignant Biliary Stricture Diagnosis

**DOI:** 10.3390/cancers14112701

**Published:** 2022-05-30

**Authors:** Mitsuru Okuno, Tsuyoshi Mukai, Keisuke Iwata, Naoki Watanabe, Takuji Tanaka, Taisei Iwasa, Kota Shimojo, Yosuke Ohashi, Akihiro Takagi, Yuki Ito, Ryuichi Tezuka, Shota Iwata, Yuhei Iwasa, Takahiro Kochi, Tomio Ogiso, Hideki Hayashi, Akihiko Sugiyama, Youichi Nishigaki, Eiichi Tomita

**Affiliations:** 1Department of Gastroenterology, Gifu Municipal Hospital, Gifu 500-8513, Japan; tsuyomukai@yahoo.co.jp (T.M.); keisukeiwata@nifty.com (K.I.); tiwasa0224@gmail.com (T.I.); kota.med.gas.ten@gmail.com (K.S.); yosuke-ohashi14@hotmail.com (Y.O.); cainable1001@yahoo.co.jp (A.T.); na9yu25@gmail.com (Y.I.); tez1101@gmail.com (R.T.); iwthalop120@yahoo.co.jp (S.I.); hey-you-asawi@hotmail.co.jp (Y.I.); kottii924@yahoo.co.jp (T.K.); tommy4000jp@yahoo.co.jp (T.O.); hide-hayashi@umin.ac.jp (H.H.); billsugiyama1025@yahoo.co.jp (A.S.); nishigak@he.mirai.ne.jp (Y.N.); etomita_jp@yahoo.co.jp (E.T.); 2Department of Gastroenterological Endoscopy, Kanazawa Medical University, Uchinada 920-0293, Japan; 3Department of Diagnostic Pathology, Gifu Municipal Hospital, Gifu 500-8513, Japan; naoki@watanabe.name (N.W.); takutt@gmhosp.gifu.gifu.jp (T.T.)

**Keywords:** malignant biliary strictures, cell block, bile cytology, bile duct biopsy, endoscopic retrograde cholangiography, immunohistochemistry

## Abstract

**Simple Summary:**

Collecting specimens for pathological diagnosis of malignant biliary stricture (MBS) is not easy. This study shows that cell block (CB) of overnight-stored bile is useful for pathological diagnosis of MBS. Cancer detectability using the CB method (62.2%) is superior to that using cytology (37.8%). When the CB method is combined with biopsy, the cancer detectability (75.6%) and accuracy rates (81.4%) are increased. In addition, immunohistochemistry can be applied to the CB method when encountering difficult cases for pathological diagnosis. The CB method utilizing overnight-stored bile can be routinely used for detecting MBS in local hospitals.

**Abstract:**

The specimen collection and subsequent pathological diagnosis of malignant biliary stricture (MBS) are difficult. This study aimed to determine whether the cell block (CB) method using overnight-stored bile is useful in the diagnosis of MBS. This trial was a single-arm prospective study involving a total of 59 patients with suspected MBS. The primary endpoint was cancer detectability and accuracy using the CB method, and a comparison with the detectability and accuracy achieved with bile cytology was made. The immunohistochemical sensitivity for maspin and p53 was also investigated in the CB and surgical specimens. We were able to collect bile from all 59 patients, and 45 of these patients were clinically diagnosed with MBS. The cancer detectability using the CB method (62.2%) was significantly higher than that using cytology (37.8%) (*p* = 0.0344). When CB was combined with biopsy, the rates of cancer detectability (75.6%) and accuracy (81.4%) increased. In eight patients who received surgical therapy, maspin- and p53-immunohistochemistry was applied to the surgical and CB specimens, and cancer cells in both specimens showed positive cytoplasmic and nuclear staining for maspin and nuclear staining for p53. The CB method is, thus, useful for detecting malignancy (UMIN000034707).

## 1. Introduction

Biliary and pancreatic cancers have the worst prognosis in Japan and the United States [1,2,3]. Biliary tract cancer (BTC) and pancreatic head cancer are known to be the major causes of malignant biliary strictures (MBS). Therefore, it is important to obtain good quality tissue samples from the MBS to obtain an accurate diagnosis and to apply an appropriate anticancer therapy. Recently, new anticancer therapies, including molecular targeted therapy or immune checkpoint inhibitor therapy, have been introduced and considered to improve prognosis [4,5,6,7,8]. A complementary diagnostic method, immunohistochemistry (IHC), is utilized for companion diagnosis to select an appropriate molecular targeted therapy [9,10,11,12].

Biopsy forceps are widely used for collecting tissue samples that are routinely used for histopathological diagnosis and IHC. However, a high level of skill is required to use biopsy forceps to pass the main papilla without endoscopic sphincterotomy (ES) and insert it into the MBS. The diagnostic sensitivity of biopsy, which is between 37% and 65.1%, is insufficient for accurate diagnosis [13,14,15,16]. Currently, endoscopic ultrasound fine-needle aspiration (EUS-FNA) has been reported to obtain adequate amounts of tissue and shows a high diagnostic sensitivity [17,18]. EUS-FNA is, thus, considered an ideal method for collecting appropriate tissues from the MBS. However, this method has a potential risk of needle tract seeding of cancer cells and peritoneal metastasis because the needle has to reach the MBS via the peritoneum [17]. A new innovative method that is easy and safe to perform and allows adequate tissue retrieval is required for an accurate pathological diagnosis of the cause of MBS.

Bile cytology after collecting bile under endoscopic transpapillary drainage is currently used for the diagnosis of patients with MBS. The sensitivity of bile cytology is low (15–58%) [19,20,21] and seems to be insufficient for diagnosis. However, the method of bile collection is simple and less invasive than biopsy and EUS-FNA, and the bile for cytology can be collected whenever required after a drainage tube is placed in the obstructed bile duct. Taken together, when the sensitivity/accuracy of the histopathological examination using bile is significantly improved, it may become a useful diagnostic tool in the clinical setting.

The first step in the cell block (CB) method is the collection of cells from effusion or needle aspiration by centrifugation to remove inflammatory cells [22]. Cells obtained from a large amount of bile are fixed in 10% neutral buffered formalin and embedded in the paraffin block. Finally, the CB method allows detailed histopathological investigation using hematoxylin and eosin (HE) staining along with IHC. However, there is only one report [20] on the application of the CB method using a small amount of bile for diagnosing the cause of MBS. To date, the effectiveness of the CB method using a large amount of bile has not been evaluated. In the current study, we aimed to evaluate the diagnostic ability and efficacy of the CB method using a large amount of overnight-stored bile for identifying the cause of MBS and compared them with the diagnostic ability and efficacy of bile cytology, which is the usual choice for cytodiagnosis.

## 2. Materials and Methods

### 2.1. Study Design

This single-arm prospective trial was conducted at Gifu Municipal Hospital, where patients clinically suspected of MBS were enrolled, and a transpapillary approach to the bile duct using endoscopic drainage was attempted (Figure 1). Informed consent was obtained from all patients. Exclusion criteria included (I) age < 20 years, (II) disease(s) already diagnosed, and (III) inability to provide informed consent. The study was approved by the Institutional Review Board of Gifu Municipal Hospital (No. 500) and registered in the University Hospital Medical Information Network Clinical Trial Registry (UMIN-CTR, UMIN000034707).

### 2.2. Patients and Procedures

This prospective study was carried out on 60 patients with biliary stricture caused by potential malignancy undergoing endoscopic retrograde cholangiography (ERC) using a standard duodenoscope (JF-260 V or TJF-260 V: Olympus Medical Systems, Tokyo, Japan) from October 2018 to October 2021 (Figure 1). Midazolam and/or pentazocine were administered intravenously for conscious sedation immediately before endoscopic retrograde cholangiopancreatography (ERCP). After ERC, a guidewire (GW) (0.025-inch VisiGlide2: Olympus Medical System Co., Ltd., Tokyo, Japan or 0.025-inch M-Through TM: Asahi Intecc Co., Ltd., Aichi, Japan) was inserted into the biliary stricture. Intraductal ultrasonography (IDUS) and ES were performed after placing the GW, if necessary. Biopsy forceps were inserted into the bile duct, and tissue samples were obtained from the biliary stricture under fluoroscopic guidance. Finally, a 5-Fr or 6-Fr endoscopic nasobiliary drainage (ENBD) tube (Gadelius Medical, Tokyo, Japan) was inserted into the obstructed bile duct over the GW (Figure 2A). The size of the ENBD tube was determined by the main operator. We immediately collected approximately 5 mL of bile from the ENBD tube and submitted it to the histopathology department for cytodiagnosis soon after the ERC procedure (the first bile cytology). After the ERC procedure, all bile samples in the bottle were stored until the morning (Figure 2B).

The overnight-stored bile samples were then submitted to the histopathology department, and CB was routinely prepared. After the overnight-stored bile was collected, multiple bile cytology (MBC) was submitted for cytodiagnosis up to five times until malignant cytodiagnosis was obtained. All adverse events (AEs) were classified according to established criteria [23].

### 2.3. Pathological Diagnosis

Tissue sections from the CB were prepared using the sodium alginate method (Figure 1) [24]. In brief, (I) bile taken from the bottle was centrifuged at 1500 rpm for 10 min (Figure 2C); (II) only the precipitate was collected and fixed in 10% neutral buffered formalin overnight; (iii) this was centrifuged again at 1500 rpm for 10 min, and the precipitate was fixed in 10% neutral buffered formalin (Figure 2D); (iv) only the precipitate was collected, and then 0.5 mL of 1% sodium alginate was added, agitated, and centrifuged again at 1500 rpm for 10 min; (v) after discarding the supernatant, 1 or 3 drops of 1 M calcium chloride solution were added; (vi) finally, it was embedded in paraffin wax (Figure 2E) and processed to make 3–4 m-thick serial sections for HE staining (Figure 2F) and IHC. For bile smear cytology with Papanicolaou staining, 2–4 glass slides for each patient were prepared. Two pathologists (N.W. and T.T.) and cytotechnologists independently made cytological and histological diagnoses, and then classified the specimens into four categories, namely “absence of atypical cells,” “non-malignant,” “suspicious for malignancy,” and “malignant.” “Suspicious for malignancy” and “malignant” were considered to be positive for cancer. The former two categories, “absence of atypical cells” and “non-malignant,” were considered negative, benign lesions.

### 2.4. Number of Malignant Cells in the CB and Biopsy Specimens

When the CB specimen was “malignant,” we counted the number of malignant cells in the CB specimen and compared the total with the number of cells present in the biopsy specimen, if available, to confirm the usefulness of the CB method.

### 2.5. Immunohistochemistry

The quality of IHC on the CB specimens was compared with that of the biopsy and surgical specimens. Primary antibodies used for IHC included maspin (Cat. No. GTX66666, GeneTex, Irvine, CA, USA) and p53 (Cat. No. M700129-2, Agilent Technologies, Santa Clara, CA, USA) antibodies. IHC was performed using an automated system (Ventana BenchMark XT system, Tucson, AZ, USA), according to the manufacturer’s instructions.

### 2.6. Final Diagnosis

The final diagnosis was made based on the histopathological diagnosis of surgical specimens and the clinical course, including disease progression and/or cancer-associated death. When the clinical course was stable for at least 6 months, the cases were considered “benign”.

### 2.7. Endpoint

The primary endpoint was cancer detectability (sensitivity), specificity, positive predictive value, negative predictive value, and accuracy for cancer achieved using the CB method and data were compared with those obtained using bile cytology. The second outcome included (I) comparison of the cancer detectability using the CB method, multiple (two or more) bile cytology, and biopsy; (II) comparison between the number of malignant cells in the CB and that in the biopsy specimens; (III) the quality of maspin- and p53-IHC of the CB specimen; and (iv) the AEs after the ERC procedure.

### 2.8. Statistical Analysis

In a previous study [20], the cancer detectability rates employing the CB method and bile cytology using less than 10 mL of bile were reported to be 53% and 28%, respectively. However, in our preliminary retrospective study of 11 cases, the values were high—91% employing the CB method using the overnight-stored bile and 55% using the first bile cytology. We, therefore, hypothesized that the cancer detectability achieved by employing the CB method using stored bile and bile cytology would be 80% and 40%, respectively. We estimated that at least 58 patients with MBS were required to obtain a bilateral significance level of 0.05 and a power of 0.9. Consequently, we planned to enroll a total of 60 patients with suspected biliary strictures in this study.

All statistical analyses of data were conducted using the JMP^®^ 15.0 software (SAS Institute Inc., Cary, NC, USA). The values included the number of patients and median (range). Fisher’s exact test was used for categorical variable analysis, and the Wilcoxon rank-sum test was used for continuous variables. Statistical significance was set at *p* < 0.05.

## 3. Results

### 3.1. Basic Characteristics of the Patients

Sixty patients were initially enrolled during the study period. The baseline characteristics of the patients are summarized in Table 1. The patients included 37 men and 23 women, with a median age of 76 years (range: 49–98 years). The Eastern cooperative oncology group performance status (ECOG-PS) score was 0 for 28 patients, 1 for 16 patients, 2 for 13 patients, and 3 for 3 patients. The median (range) value for alanine aminotransferase (AST) and aspartate aminotransferase (ALT) activity was 89 (13–540) and 119 (9–1246) IU/L, respectively; the total bilirubin level was 1.5 (0.3–34.2) mg/dL; the white blood count was 6490 (2890–20750)/µL; the neutrophil count was 4150 (1030–19090)/µL; and hemoglobin level was 12.0 (8.7–15.8) g/dL. The final diagnoses of malignancy were made in 45 patients (75.0%), which included 26 extrahepatic cholangiocarcinomas (43.3%), 7 gallbladder cancers (11.7%), 5 ampullary cancers (8.3%), 4 pancreatic head cancers (6.7%), 2 intrahepatic cholangiocarcinomas (3.3%), and 1 hepatocellular carcinoma (1.7%). A total of 14 benign diseases included 4 cases (6.7%) of primary sclerosing cholangitis; 4 cases (6.7%) of IgG4-related sclerosing cholangitis; 1 case (1.7%) of bile duct hamartoma; 1 case (1.7%) of liver cirrhosis; 1 case (1.7%) of calculous cholecystitis; and 3 cases (5.0%) of other inflammatory and non-malignant stenosis (Figure 3). Among the cases enrolled, one patient who had intrahepatic biliary stricture did not receive the final diagnosis because we could not follow up with the patients. Therefore, we analyzed a total of 59 cases (Figure 1 and Figure 3, Table 1). Among them, 16 patients with cholangiocarcinoma, gallbladder cancer, or ampullary cancer received surgical therapy, and their surgical specimens were available for pathological analyses (Figure 3).

### 3.2. Endoscopic Procedure and Findings

ERC was performed on 60 patients suspected of having MBS to obtain bile and tissue samples (Figure 1). We were able to access the bile ducts of all patients and path through the biliary stricture with GW. ES was performed in 46 patients (76.7%). While tissue sampling using biopsy forceps was performed on 47 patients, we were unable to obtain biopsy samples from 13 patients. In six cases, biopsy forceps could not reach the biliary stricture caused by malignancy, such as intrahepatic cholangiocarcinoma. There were six cases in which only ENBD tube placement was performed due to cholangitis and one case in which tissue collection by EUS-FNA was attempted. 5 Fr ENBD tube was placed in an obstructed bile duct in 32 patients, and a 6 Fr ENBD tube was placed in 28 patients. Under the GW assistant, we inserted the ENBD tube into the gallbladder of two patients suspected of having gallbladder cancer. We obtained the bile of all patients for the first bile cytology and the CB method after ERC. The mean volume of bile for the CB method was 180 mL (range: 40–300 mL). MBC was performed two times on an average (range: 2–5 times). One patient who underwent ERC and ENBD placement to clarify the cause of intrahepatic bile duct stricture could not be followed-up considering the patient’s will.

### 3.3. Cancer Detectability

As summarized in Table 2, the cancer detectability (sensitivity) of the CB method was 62.2% (28 of 45 malignant cases), and the first cytology was 37.8% (17 of 45 malignant cases). The efficacy of the CB method in determining malignancy was significantly higher than that of the first cytology (*p* = 0.0344). For each malignant lesion, the sensitivity of BTC (67.5%, 27 of 40 cases, *p* = 0.0243) using the CB method was significantly greater than that (40%, 16 of 40 cases) using the first cytology.

We performed MBC two times on an average (range: 2–5 times). Biopsy was performed three times on an average (range: 2–6 times) for 47 suspicious cases of MBS. The sensitivity of MBC and biopsy was 60% and 65.8%, respectively, and the difference between the two methods was insignificant.

Table 3 summarizes the diagnostic efficacies of the CB method, first cytology, MBC, and biopsy results. Although the specificity and positive predictive value were 100% for all the methods, negative predictive values were 45.2%, 33.3%, 43.8%, and 40.9%, respectively, and the accuracy rates were 71.2%, 52.5%, 69.5%, and 72.3%, respectively. The difference in the accuracy rate between the CB method and the first cytology was not significant (*p* = 0.0575). When the CB method was combined with biopsy, the rates of sensitivity (75.6%, 34 of 45 cases) and accuracy (81.4%, 48 of 59 cases) increased. Among 45 patients with a final diagnosis of MBS, the CB method led to a correct diagnosis in 28 patients, whereas it failed in 17. Biopsy was added for a correct diagnosis in 6 of the 17 failed cases (35.3%) (Table 4).

### 3.4. Malignant Cell Count

We observed malignant epithelial cells in both the CB and biopsy specimens (Figure 4A). As shown in Figure 4B, the mean number of malignant cells on a glass slide made using the CB method was 70 (range: 10–1110 cells), whereas the value on a biopsy specimen that included a mean of three small tissues (range: 2–6 tissues) was 110 cells (range: 30–280 cells). There was no significant difference in the number of malignant cells per glass slide between the CB and biopsy specimens (*p* = 0.17).

### 3.5. Immunohistochemistry

Eight of sixteen patients who underwent surgical therapy were diagnosed with MBS using the CB specimen. They included BTC (four patients with extrahepatic cholangiocarcinoma, two with gallbladder cancer, one with intrahepatic cholangiocarcinoma, and one with ampullary cancer). Five cases had negative biopsy results; therefore, biopsy specimens were available from only three of these eight cases.

Surgical specimens obtained from eight patients were processed for IHC using antibodies against maspin and p53 (Figure 5). While positive expression of maspin and p53 was found in all eight patients, the intensity of the IHC staining differed considerably among the cases. Maspin and p53 immunoreactivity were further semi-quantitatively evaluated using a modified labeling index (LI) system. Briefly, the percentage of immunoreactivity LI was categorized as score 0 (no expression), score 1+ (up to 10%), score 2+ (11–50%), and score 3+ (51–100%). While relatively weak positive stainability of maspin was observed in the cytoplasm and/or nucleus of cancer cells in the surgical and biopsy specimens, strong stainability of maspin was observed in the CB specimens. Immunohistochemical stainability of maspin in the surgical, CB, and biopsy specimens was similar. Positive reactivity of p53 was observed sparsely in the nucleus of cancer cells in the surgical specimens (Figure 5). Therefore, the CB and biopsy specimens sometimes contained cells that were not stained, even though the cancer cells were collected. In addition, the case of score 1+ surgical specimen could not be confirmed using the CB specimen (Table 5).

### 3.6. Adverse Events

Post-ERCP pancreatitis (PEP) was observed in four of sixty patients (6.7%). All these cases had mild PEP, and the size of the ENBD tube used was 5-Fr in three patients and 6-Fr in one patient. No other AEs or subsequent deaths occurred.

## 4. Discussion

MBS is usually caused by pancreatic cancer and/or BTC. EUS-FNA, which has a high accuracy rate of diagnosis and fewer AEs, is one of the standard methods for clinical diagnosis of pancreatic cancer. On the contrary, diagnosis of BTC is sometimes difficult because of the low sensitivity of both bile cytology and biopsy of MBS. The benefit of bile cytology is that bile can be obtained easily when compared with the difficulty in inserting the biopsy forceps from regions, including the gallbladder or distal intrahepatic bile duct. Therefore, if a high-sensitivity diagnostic specimen can be made from bile, it may improve MBS diagnosis. The CB method, which is based on cytological materials, has been reported to improve histopathological diagnosis and is effective for determining molecular biomarkers. This led to novel therapies based on genomic medicine in pancreatic/biliary malignancy as well as cancers in other tissues, including lung, ovaries, and peritoneum [25,26,27,28]. To make CB specimens for determining the cause(s) of biliary stricture, a large amount of bile using the ENBD tube from obstructed bile duct is needed. ENBD tube placement is safer and less invasive than percutaneous transhepatic biliary drainage and is easier to perform than transpupillary biopsy.

Noda et al. [20] previously reported the efficacy of the CB method with a small amount of bile for the diagnosis of bile duct cancer. They collected bile through a catheter with negative pressure using a 10 mL syringe during the ERC procedure. They reported that the sensitivity (52.9%) of the CB method was significantly higher than that (27.9%) of smear cytology (*p* = 0.014). In the current study, a large amount of bile was used to prepare the CB specimen, resulting in high diagnosis sensitivity (62.2%). We also evaluated whether there is a relationship between the bile volume and diagnosis of malignancy using the CB method in 45 malignant cases. In the receiver operating characteristic (ROC) curve analysis of bile volume and diagnosis of malignancy, the area under the curve (AUC) was 0.61. The optimal cutoff was calculated to be 160 mL, with a sensitivity of 43% and specificity of 82% for the diagnosis of malignancy (Figure 6). When the cutoff value was set at 160 mL, there was no significant difference between the high (≥160 mL) and low (<160 mL) volume bile groups (*p* = 0.28). The reason why there was no significant difference is believed to be the fact that very few cases were examined employing the cell block method using a small amount of bile because overnight-stored bile was used.

The CB specimen contained a sufficient number (mean number: 70 cells/specimen, *p* = 0.17) of malignant cells, equivalent to that (mean number: 110 cells/specimen) in the biopsy sample. On average, MBC needed to be repeated two times for an essential diagnosis. Although this seems low for MBC, the diagnosis of 44.4% (20/45) cases was already made using the CB method before performing the third bile cytology. Thus, the high sensitivity of the CB method can reduce the number of unnecessary bile cytology repetitions. In this study, the CB method, combined with biopsy, improved the diagnostic performance. The CB method can be used to evaluate the bile collected from the region where biopsy forceps cannot be inserted, resulting in high sensitivity (75.6%) of diagnosis. Therefore, the combination of the CB method and biopsy may increase the diagnostic sensitivity. Recently, liquid biopsy using blood and next-generation sequencing (NGS) of brushing cytological specimens or bile from BTC patients was reported to have a high sensitivity (73–100%) and to be effective for evaluating DNA in cancer cells [29,30,31,32]. However, in everyday clinical practice in front-line hospitals, it is important to consider the time and cost required to ensure an accurate diagnosis. In Japan, the use of NGS is limited to patients who have already been diagnosed with malignant disease and for whom standard therapies have become ineffective.

The major advantages associated with the use of the CB method are that even though it is simple and inexpensive, it provides appropriate pathological diagnosis. Recently, the measures of microsatellite instability (MSI) and neurotrophic receptor tyrosine kinase (NTRK) expression have been important companion diagnostic tests for determining treatment strategies. Such companion diagnoses can be evaluated by IHC and/or NGS to identify genomic alterations in malignant tissues. BTC is one of the cancer types where it is difficult to diagnose because of insufficient amounts of samples obtained; therefore, it is difficult to perform the cancer gene panel test using NGS. However, the efficacy of PD-1 inhibitors was reported in patients who were deficient in DNA mismatch repair genes, as demonstrated using IHC alone [5]. In addition, NTRK expression can be evaluated using IHC as a screening test [9,10,11,12]. When compared to NGS, IHC can be used to rapidly evaluate the MSI status and NTRK expression in smaller specimens. Therefore, when the quality of CB and surgical specimens is the same, it is considered that the CB specimen can be potentially used as a sample for companion diagnostics and provides an alternative when NGS testing is difficult.

In this study, all patients who underwent surgical treatment had BTC. The stainability of maspin and p53 was evaluated using IHC. Cytoplasmic and nuclear maspin staining was uniformly observed in malignant cells in the surgical (8/8 cases, 100%) and CB (8/8 cases, 100%) specimens, with slightly different intensities, suggesting that the CB specimens can potentially be used as samples for IHC companion diagnostics. Although only three of eight biopsy specimens were available for maspin-IHC, their staining properties were similar to those of surgical and CB specimens. On the contrary, positive reactivity of p53 was sparsely observed only in the nuclei of cancer cells in the surgical specimens. When IHC staining was weak and sparse, as in the case of p53 score 1+ staining in surgical specimens (Table 4), it was thought that it might be difficult to confirm the evaluation of IHC staining because the stainable cells were not collected. Although it is difficult to draw conclusions due to the small number of cases, it is important to note that the results of the evaluation of CB and biopsy specimens using IHC were similar to those for the surgical specimens. The current study used overnight-stored bile, which is a large amount of bile (mean volume: 180 mL); however, this method potentially lowered the quality of malignant cells, which may be caused by overnight storage of bile. The reason why a large number of high-quality malignant cells were present in the CB specimen from overnight-stored bile might be the fact that centrifugation could exclude degenerated and necrotic debris (Figure 2A). These findings also suggest that the CB method can be used as an alternative to biopsy specimens when IHC is required in cases in which obtaining biopsy specimens is difficult. In the current study, we did not perform a gene panel test using CB specimens. However, we were able to obtain a large number of malignant cells in the CB specimens, suggesting that the testing of genetic alterations is possible, as reported in cases of lung cancer [33,34]. It is necessary to evaluate whether the CB samples would be suitable for NGS in a future study.

The occurrence of AE was 6.7%, all of which were mild PEPs. The frequency of PEP in BTC cases has been reported to be 0–9.6%, and the dilated bile duct is a risk factor for PEP [15,20,35,36,37]. Considering that our study included a large number of BTC with dilated bile ducts, the incidence (6.7%) of PEP was acceptable. The placement of the ENBD tube may be one of the causes of obstruction of the pancreatic duct. The different diameters (5Fr/6Fr) of the ENBD tubes did not influence the occurrence of PEP and cancer detection rate.

Our study has several limitations. First, this study included a widely varied patient population from a single institute, and the total number of patients analyzed was small (59 cases of suspicious MBS and 45 cases of malignancy). The sample size was also small, and only Asian people were enrolled. However, the CB method showed a significantly higher sensitivity rate (62.2%) for detecting malignancy in 45 cases than single bile cytology (37.8%). Second, on average, two iterations of MBC are too small because MBC stopped when the malignancy was detected by cytology or the CB method. Third, although the usefulness of IHC has been confirmed, NGS analysis using CB specimens was not evaluated. Whether CB samples can be used for evaluation employing NGS is a topic for future studies. Finally, we assessed IHC in BTC but not in other cancers, such as pancreatic cancer. In general, MBS is caused by pancreatic cancer or BTC. Pancreatic cancer is usually diagnosed using EUS-FNA before ERCP in our hospital; therefore, in the current study, patients who had already undergone EUS-FNA and had a diagnosis of pancreatic cancer were not enrolled.

## 5. Conclusions

Our study demonstrates that higher cancer detectability was achieved with the CB method, which allows for a large specimen volume, than with bile cytology. The specimens prepared using the CB method were of sufficient quality for processing IHC, an important tool for further investigations. The CB specimen requires only ENBD placement and has a low risk of AEs. Thus, the CB method is an easy, functional, and safe method for the collection of histopathological specimens.

## Figures and Tables

**Figure 1 cancers-14-02701-f001:**
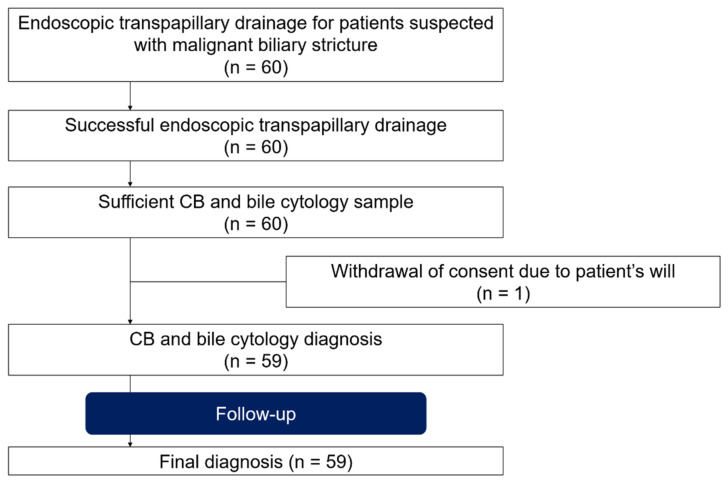
Study patient flow.

**Figure 2 cancers-14-02701-f002:**
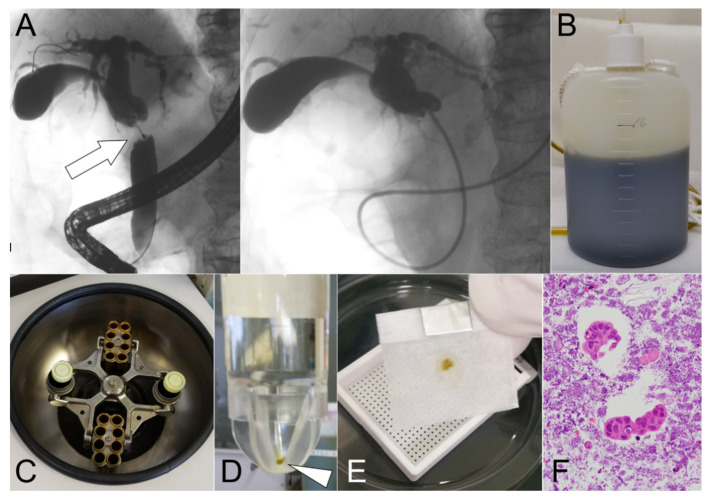
The cell block method was performed using overnight-stored bile. (**A**) Suspicious malignant biliary stricture (MBS) was detected in the middle bile duct (arrow). Endoscopic nasobiliary drainage (ENBD) tube was placed in the occluded bile duct. (**B**) Bile from the ENBD tube was stored overnight after ENBD tube placement. (**C**) The stored bile was processed by centrifugation for 10 min. (**D**) Only the precipitate (arrowhead) was collected and fixed in 10% neutral buffered formalin. (**E**) The precipitate was then embedded in paraffin wax. (**F**) The malignant specimen could be detected by hematoxylin and eosin staining of the section.

**Figure 3 cancers-14-02701-f003:**
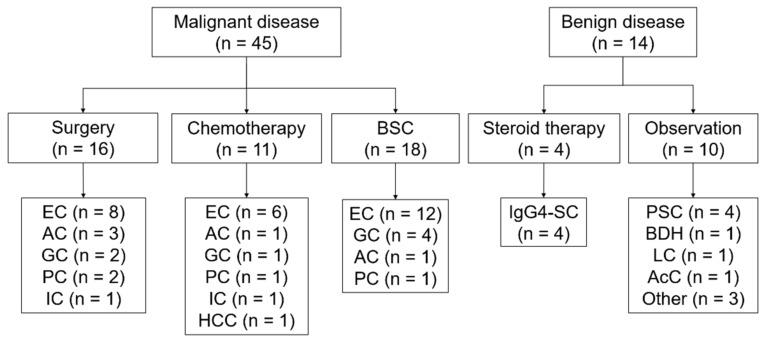
Outline of the process used to reach the final diagnosis. BSC, best supportive care; EC, extrahepatic cholangiocarcinoma; AC, ampullary cancer; GC, gallbladder cancer; PC, pancreatic cancer; IC, intrahepatic cholangiocarcinoma; HCC, hepatic cell carcinoma; IgG4-SC, IgG4-related sclerosing cholangitis; PSC, primary sclerosing cholangitis; BDH, bile duct hamartoma; LC, liver cirrhosis; and AcC, acalculous cholecystitis.

**Figure 4 cancers-14-02701-f004:**
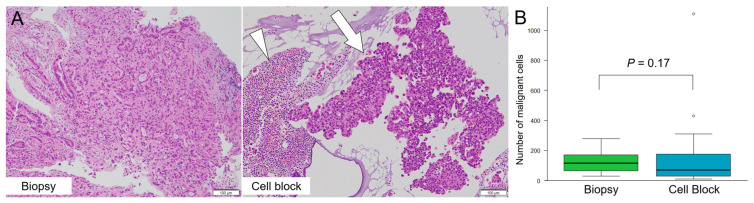
The comparison of malignant cell count on the cell block (CB) and biopsy specimens. (**A**) Degenerated cells (arrowhead) and cancer cells cluster (arrow) are easily identifiable on a CB specimen using overnight-stored bile after centrifugation. (**B**) The mean number of malignant cells on a glass slide with the CB and biopsy specimens was 70 (range: 10–1110) and 110 (range: 30–280), respectively. There is no significant difference between the CB method and biopsy specimens (*p* = 0.17).

**Figure 5 cancers-14-02701-f005:**
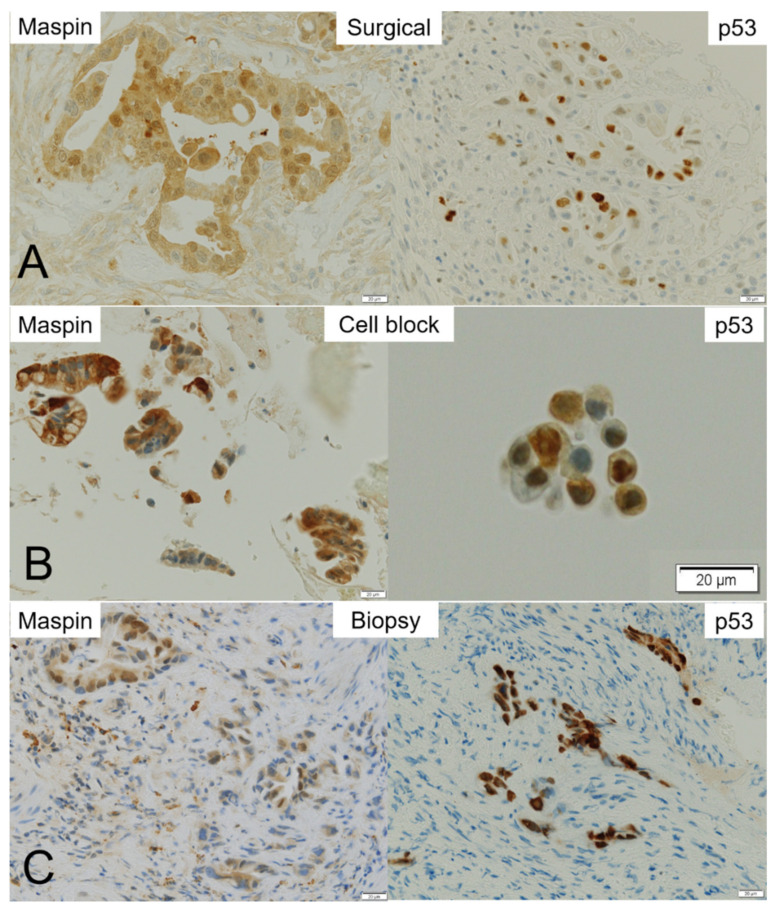
Representative maspin- and p53-immunohistochemistry in the surgical, cell block (CB), and biopsy specimens. Immunohistochemistry of maspin and p53 in (**A**) surgical, (**B**) CB, and (**C**) biopsy specimens. While maspin staining is found in almost all malignant cells in the surgical specimen, p53 staining is sparse in the nucleus of cancer cells. (**A**) Although nuclear positivity of maspin is clearly observed in adenocarcinoma cells, p53 expression is weak in their cytoplasm and sparse in the nucleus of cancer cells. (**B**) Maspin is strongly positive in the cytoplasm of adenocarcinoma cells in the CB specimen. On the contrary, the cancer cell nuclei are sparsely stained with p53. (**C**) Maspin staining is positive in the cytoplasm and/or nuclei of adenocarcinoma cells in a biopsy specimen. p53 staining is positive in the nuclei of cancer cells.

**Figure 6 cancers-14-02701-f006:**
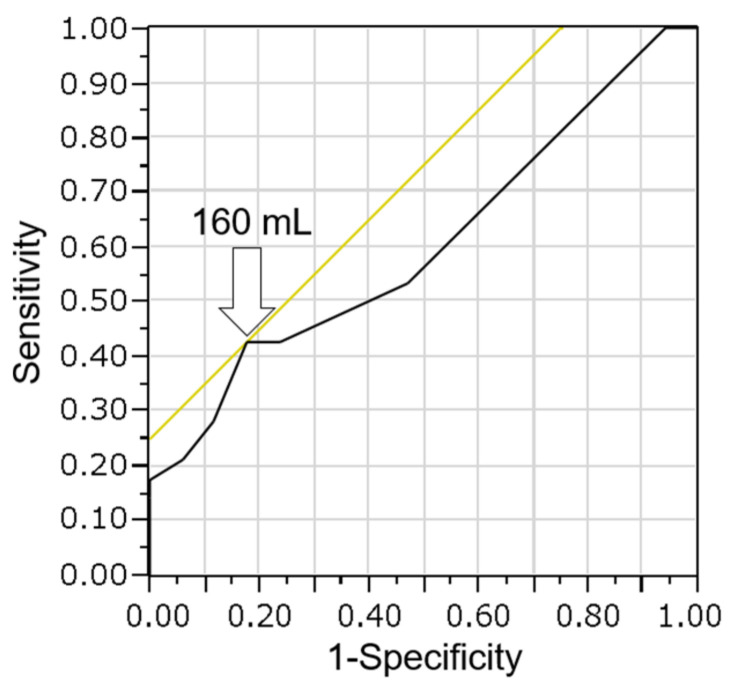
Receiver operating characteristic curve analysis of bile volume and diagnosis of malignancy using the cell block method. The optimal cutoff was calculated as 160 mL (arrow), with a sensitivity of 43% and a specificity of 82%. The area under the curve is 0.61.

**Table 1 cancers-14-02701-t001:** Patients’ baseline characteristics.

	*n* = 60
Sex, female/Male *n* (%)	23 (38.3%)/37 (61.7%)
Age (years), median (range)	76 (49–98)
Endoscopic sphincterotomy	46 (76.7%)
Diameter of ENBD tube 5 Fr:6 Fr (number)	32:28
Eastern cooperative oncology group performance status (ECOG-PS), *n* (%)	
0	28
1	16
2	13
3	3
Alanine aminotransferase (AST), median (range), IU/L	89 (13–540)
Aspartate aminotransferase (ALT), median (range), IU/L	119 (9–1246)
Total bilirubin level, median (range), mg/dL	1.5 (0.3–34.2)
White blood count, median (range)/µL	6490 (2890–20750)
Neutrophils, median (range)/µL	4150 (1030–19090)
Hemoglobin, median (range), g/dL	12.0 (8.7–15.8)
Location of stricture in the bile duct	
Intrahepatic	6 (10.0%)
Portal	16 (26.7%)
Superior	17 (28.3%)
Inferior	21 (35.0%)
Type of malignancy, *n* (%)	
Extrahepatic cholangiocarcinoma	26 (43.4%)
Gallbladder cancer	7 (11.7%)
Ampullary cancer	5 (8.4%)
Pancreatic head cancer	4 (6.6%)
Intrahepatic cholangiocarcinoma	2 (3.3%)
Hepatocellular carcinoma	1 (1.7%)
Benign, *n* (%)	
Primary sclerosing cholangitis	4 (6.6%)
IgG4-related sclerosing cholangitis	4 (6.6%)
Bile duct hamartoma	1 (1.7%)
Liver cirrhosis	1 (1.7%)
Acalculous cholecystitis	1 (1.7%)
Other inflammatory non-malignant stenosis	3 (5.0%)
Withdrawal case due to patient’s will	1 (1.7%)

**Table 2 cancers-14-02701-t002:** Comparison of cancer detectability using the cell block (CB) method and bile cytology.

Overall (*n* = 59)	CB Method, *n* (%)	Bile Cytology, *n* (%)	*p*-Value
Malignant (*n* = 45)	28 (62.2%)	17 (37.8%)	0.0344
Extrahepatic cholangiocarcinoma (*n* = 26)	15 (57.7%)	12 (46.2%)	0.58
Gallbladder cancer (*n* = 7)	7 (100%)	2 (28.6%)	0.0210
Ampullary cancer (*n* = 5)	3 (60.0%)	1 (20.0%)	0.52
Pancreatic head cancer (*n* = 4)	1 (25.0%)	1 (25.0%)	1.0
Intrahepatic cholangiocarcinoma (*n* = 2)	2 (100%)	1 (50.0%)	1.0
Hepatic cell carcinoma (*n* = 1)	0 (0%)	0 (0%)	1.0
Benign (*n* = 14)	14 (100%)	14 (100%)	1.0

**Table 3 cancers-14-02701-t003:** Diagnostic efficacy of the cell block (CB) method, bile cytology, and biopsy.

	CB Method	First Bile Cytology	Multiple Bile Cytology	Biopsy
Sensitivity	62.2% (28/45)	37.8% (17/45) *	60.0% (27/45)	65.8% (25/38)
Specificity	100% (14/14)	100% (14/14)	100% (14/14)	100% (9/9)
PPV	100% (28/28)	100% (17/17)	100% (27/27)	100% (25/25)
NPV	45.2% (14/31)	33.3% (14/42)	43.8% (14/32)	40.9% (9/22)
Accuracy	71.2% (42/59)	52.5% (31/59) ^†^	69.5% (41/59)	72.3% (34/47)

** p* < 0.05, compared with the CB method (Fisher’s test). † *p* = 0.0575, compared with the CB method (Fisher’s test).

**Table 4 cancers-14-02701-t004:** Comparison of diagnosis using the cell block (CB) method and biopsy in malignant disease.

	CB Method	
	Malignant (*n* = 28)	False Negative (*n* = 17)
Biopsy		
Malignant (*n* = 25)	19	6
False negative (*n* = 13)	7	6
Unperformed case (*n* = 7)	2	5

**Table 5 cancers-14-02701-t005:** Comparison of the maspin/p53-immunohistochemistry (IHC) in each specimen.

Case	Surgical Specimen (Maspin/p53)	CB Method (Maspin/p53)	Biopsy (Maspin/p53)
1	2 + /1+	1 + /0	2 + /1+
2	2 + /1+	3 + /1+	3 + /1+
3	2 + /2+	3 + /2+	2 + /3+
4	2 + /1+	2 + /0	N.A.
5	1 + /1+	1 + /0	N.A.
6	3 + /1+	3 + /2+	N.A.
7	3 + /2+	2 + /3+	N.A.
8	3 + /1+	2 + /2+	N.A.

IHC staining was scored from 0 to 3+ based on intensity and positive rate. N.A.: not available.

## Data Availability

Not applicable.

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
