# Peer review of "Evaluation of the Cell Block Method Using Overnight-Stored Bile for Malignant Biliary Stricture Diagnosis"

_cancers, 2022, doi:10.3390/cancers14112701_

Round 1
Reviewer 1 Report
The article of Okuno et al is of interest since biliary tract cancer (BTC) has a very bad prognosis and is known to be the major cause of malignant biliary strictures (MBS). Therefore, it is important to obtain good quality tissue samples from the MBS to obtain accurate diagnosis and apply appropriate anti-cancer therapy. This study has demonstrated that cell block of overnight-stored bile is useful for pathological diagnosis of MBS, because the cancer detectability is superior to that of cytology and when is combined with biopsy, the detectability and accuracy rates are increased. The article is well written and structured, the methodology is appropriate, and results are well presented and are in concordance with the discussion. Moreover, bibliography is actual and supported the state of the art. The conclusion is interesting and could help in clinical practice to dramatically improve the diagnosis of BTC. I would like to recommend this manuscript to be published in the present form after authors check minor spelling and English grammar of the wok.
Author Response
Response:
Many thanks for your warm comments that encouraged us to further investigation/research of oncogenesis. We have revised the manuscript and checked English by a native speaker.

Reviewer 2 Report
The manuscript by Okuno et al. describes a modification of the cell block method for the diagnosis of malignant biliary strictures.
The novelty of the results presented by Okuno et al. is only incremental as the cell block method was already described several years ago (Sakamoto et al. Open Journal of Pathology 2012). Moreover, the sensitivity reached (62.2%) upon overnight-storing bile, is far from other methods recently published regarding sequencing of cellular brushing (Singhi et al Gut 2020) or bile cell free DNA (Arechederra et al Gut 2022), with sensitivities of 73% and 100% respectively for the diagnosis of malignant stenoses.
Moreover Okuno et al. do not clearly describe the volume of overnight-stored bile used in each case, and whether there is a relationship between the volume used and the sensitivity of the method.
Author Response
Response: Many thanks for new information regarding sequencing of cellular brushing (PMID: 30971436) and bile acid free DNA (PMID: 34285068) and your kind suggestion regarding the volume of overnight-stored bile acid used, Accordingly, we have added the sentences citing two important papers. Also, information about the volume of overnight-stored bile acid you indicated has been added.
Lines 317-326: We also evaluated whether there is a relationship between the bile volume and diagnosis of malignancy using the CB method in 45 malignant cases. In the receiver operating characteristic (ROC) curve analysis of bile volume and diagnosis of malignancy, the area under the curve (AUC) was 0.61. The optimal cutoff was calculated to be 160 mL, with a sensitivity of 43% and specificity of 82% for the diagnosis of malignancy (Figure 6). When the cutoff value was set at 160 mL, there was no significant difference for the high (≥160 mL) and low (<160 mL) volume bile groups (P = 0.28). The reason why there was no significant difference is believed to be the fact that very few cases were examined employing the cell block method using a small amount of bile because overnight-stored bile was used.
Lines 341-349: Recently, liquid biopsy using blood, and next-generation sequencing (NGS) of brushing cytological specimens or bile from BTC patients was reported to have a high sensitivity (73–100%), and to be effective for evaluating DNA in cancer cells [29-32]. However, in everyday clinical practice in front line hospitals, it is important to consider the time and cost required to ensure accurate diagnosis. In Japan, the use of NGS is limited to patients who have already been diagnosed with malignant disease and for whom standard therapies have become ineffective.
The major advantages associated with the use of the CB method are that even though it is simple and inexpensive, it provides appropriate pathological diagnosis.
Lines 383-387: In the current study, we did not perform a gene panel test using CB specimens. However, we were able to obtain a large number of malignant cells in the CB specimens, suggesting that the testing of genetic alterations is possible, as reported in cases of lung cancer [33, 34]. It is necessary to evaluate whether the CB samples would be suitable for NGS in a future study.
Lines 401-404: Third, although the usefulness of IHC has been confirmed, NGS analysis using CB specimens was not evaluated. Whether CB samples can be used for evaluation employing NGS is a topic for future studies.

Reviewer 3 Report
Dear Editor, thank you so much for inviting me to revise this manuscript.
Understanding the role of these approaches in this setting is a mandatory need and the study aimed to determine whether the cell block (CB) method using 19 overnight-stored bile is utilized for the diagnosis of MBS.
On the basis of the above, it addresses a current topic.
The manuscript is quite well written and organized. English should be improved.
Figures and tables are comprehensive and clear. However, as you could see below, some points should be elucidated.
We suggest the following modifications:
- Introduction section: although the authors correctly included important papers in this setting, we believe a couple of studies regarding the diagnosis of cholangiocarcinoma and biliary tract cancer should be cited within the introduction ( PMID: 24581682 ; PMID: 33611090 ; PMID: 28994423; PMID: 32111744) only for a matter of consistency. We think it might be useful to introduce the topic of this study.
- In addition, we believe some issues deserve further discussion. In everyday clinical practice, we know that the pathologic confirmation of diagnosis is necessary before any non-surgical treatment and can be challenging in BTC, particularly in patients affected by primary sclerosing cholangitis and biliary strictures. In fact, decisions to undertake biopsies should follow a multidisciplinary discussion, especially in potentially resectable tumors. Moreover, endoscopic imaging and tissue sampling are useful but, sadly, biopsy samples are often inadequate for molecular profiling, and in addition, tissue sampling has reported high specificity but low sensitivity in diagnosis of malignant biliary strictures. Finally, the highly desmoplastic nature of BTC limits the accuracy of cytological and pathological approaches.
On the basis of these premises, in this scenario, it is urgent to develop new strategies in order to anticipate the diagnosis identifying BTC at an early, resectable stage, and to obtain sufficient material with which to perform genomic analysis. Among these strategies, liquid biopsy has received growing attention over the years, given the promising applications in cancer patients. More specifically, several studies have shown the potential role of liquid biopsy, and the authors should discuss this point, also reporting recent studies in this setting (doi: 10.3390/cells9030721; doi: 10.21873/cgp.20203).
- Methods and Statistical Analysis: nothing to add.
- Table 1, Baseline characteristics of study participants. The authors should report some important parameters, including GGT levels (gamma glutamyl transferase), eastern cooperative oncology group (ECOG) performance status (PS), white blood count, neutrophils, hemoglobin, since some of these biochemical parameters have been suggested as independent prognostic factors for survival in biliary tract cancer patients.
In particular, the most relevant independent prognostic factor is probably ECOG-PS, an extremely simple clinical parameter which may help to guide therapeutical choices.
For example, although based on a different setting, all international guidelines suggest that chemotherapy with a single-agent should be preferred in BTC patients with ECOG-PS 2, also on the basis of a meta-analysis including the ABC-02 and the BT-22 trials, showing that BTCs with poor ECOG-PS do not seem to derive benefit from the reference doublet, and thus suggesting that gemcitabine monotherapy is a feasible treatment in this patient population.
- Discussion section: Interesting section.
However, some changes and some additions are necessary.
Of note, the authors should expand the Discussion section, including a more personal perspective to reflect on. For example, they could answer to the following questions – in order to facilitate the understanding of this complex topic to readers: what potential do this study hold? What are the knowledge gaps and how do researchers tackle them? How do you see this area unfolding in the next 5 years?
We think it would be extremely interesting for the readers, where novel treatment options are opening the doors of a new world, with the hope to lower the recurrence rates of these aggressive malignancies.
One additional little flaw: the authors should better explain the limitations of their work, in the last part of the Discussion.
We believe this article is suitable for publication in the journal although major revisions are needed. The main strengths of this paper are that it addresses an interesting and very timely question and provides a clear answer, with some limitations.
Certainly, the study is limited to an Asian population with a very small sample size, and authors should further express this point.
Second, the study included a widely varied patient population from a single institute and the total number of patients analyzed was relatively small. Finally, the authors should report some baseline characteristics of patients which have not been included.
We suggest a linguistic revision, the addition of some references for a matter of consistency and some clarifications and extensive changes regarding some crucial points in everyday clinical practice of biliary tract cancers.
Author Response
Response: We would like to thank the reviewer for your constructive suggestions on how to improve the quality of this manuscript. In the revised version, we have addressed the concerns and suggestions you indicated.
-As you recommended, we have cited 4 important papers (PMID: 24581682; PMID: 33611090; PMID: 28994423; PMID: 32111744).
Lines 341-349: Recently, liquid biopsy using blood, and next-generation sequencing (NGS) of brushing cytological specimens or bile from BTC patients was reported to have a high sensitivity (73–100%), and to be effective for evaluating DNA in cancer cells [29-32]. However, in everyday clinical practice in front line hospitals, it is important to consider the time and cost required to ensure accurate diagnosis. In Japan, the use of NGS is limited to patients who have already been diagnosed with malignant disease and for whom standard therapies have become ineffective.
The major advantages associated with the use of the CB method are that even though it is simple and inexpensive, it provides appropriate pathological diagnosis.
Lines 401-404: Third, although the usefulness of IHC has been confirmed, NGS analysis using CB specimens was not evaluated. Whether CB samples can be used for evaluation employing NGS is a topic for future studies.
-In the discussion section, we had added several sentences by citing two important papers (PMID: 32183400 and PMID: 32859625), as you suggested.
-Table 1: As you recommended, we have added ECOG-PS, several biochemical and blood data (AST, ALT, WBC, Neutrophils, Hemoglobin) in Table 1 and text. Instead of GGT, we usually measured AST, ALT, total bilirubin.
Line 179-185: The Eastern cooperative oncology group performance status (ECOG-PS) score was 0 for 28 patients, 1 for 16 patients, 2 for 13 patients, and 3 for 3 patients. The median (range) value for alanine aminotransferase (AST) and aspartate aminotransferase (ALT) activity was 89 (13–540) and 119 (9–1246) IU/L, respectively; the total bilirubin level was 1.5 (0.3–34.2) mg/dL; the white blood count was 6490 (2890–20750)/µL and neutrophil count was 4150 (1030–19090)/µL; and hemoglobin level was 12.0 (8.7–15.8) g/dL.
-Discussion section: As you suggested, we have added several sentences regarding limitations of our study and future studies in our hospital.
Lines 383-387: In the current study, we did not perform a gene panel test using CB specimens. However, we were able to obtain a large number of malignant cells in the CB specimens, suggesting that the testing of genetic alterations is possible, as reported in cases of lung cancer [33, 34]. It is necessary to evaluate whether the CB samples would be suitable for NGS in a future study.
Line 395-398: First, this study included a widely varied patient population from a single institute and the total number of patients analyzed was small (59 cases of suspicious MBS and 45 cases of malignancy). The sample size was also small, and only Asian people were enrolled.
Lines 401-404: Third, although the usefulness of IHC has been confirmed, NGS analysis using CB specimens was not evaluated. Whether CB samples can be used for evaluation employing NGS is a topic for future studies.

Round 2
Reviewer 2 Report
Thank you.
Reviewer 3 Report
Acceptance.